# Barriers and Facilitators to Hepatitis C Virus (HCV) Treatment for Aboriginal and Torres Strait Islander Peoples in Rural South Australia: A Service Providers’ Perspective

**DOI:** 10.3390/ijerph20054415

**Published:** 2023-03-01

**Authors:** David Lim, Emily Phillips, Clare Bradley, James Ward

**Affiliations:** 1Translational Health Research Institute, School of Health Sciences, Campbelltown, NSW 2560, Australia; 2National Aboriginal Community Controlled Health Organisation, Canberra, ACT 2601, Australia; 3Poche Centre for Indigenous Health, The University of Queensland, Toowong, QLD 4066, Australia

**Keywords:** hepatitis C, rural, remote, Indigenous peoples, primary healthcare, health services

## Abstract

This study explored the barriers and facilitators to hepatitis C virus (HCV) treatment for Aboriginal and Torres Strait Islander peoples in rural South Australia as viewed from a healthcare provider perspective in the era of direct acting antivirals (DAAs). Phase 1 was a qualitative systematic review examining the barriers and enablers to diagnosis and treatment amongst Indigenous peoples living with HCV worldwide. Phase 2 was a qualitative descriptive study with healthcare workers from six de-identified rural and regional Aboriginal Community-Controlled Health Services in South Australia. The results from both methods were integrated at the analysis phase to understand how HCV treatment could be improved for rural Aboriginal and Torres Strait Islander peoples. Five main themes emerged: the importance of HCV education, recognizing competing social and cultural demands, the impact of holistic care delivery and client experience, the effect of internal barriers, and overlapping stigma, discrimination, and shame determine how Indigenous peoples navigate the healthcare system and their decision to engage in HCV care. Continued efforts to facilitate the uptake of DAA medications by Aboriginal and Torres Strait peoples in rural areas should utilize a multifaceted approach incorporating education to community and cultural awareness to reduce stigma and discrimination.

## 1. Introduction

Hepatitis C virus (HCV) infection is a highly stigmatized condition and currently infects 58 million people, with about 1.5 million new infections per year [1]. Approximately 70% of people infected with HCV will develop chronic infection, which increases the risk of developing cirrhosis and liver cancer [2,3,4,5]. In Australia, HCV remains one of the most commonly notified diseases, with Aboriginal and Torres Strait Islander peoples (particularly in rural and remote areas) recognized as a high-risk population group [6]. There is no vaccine; however, current antiviral treatments can cure >95% of those infected [1]. In light of the effectiveness of treatment, the World Health Organization (WHO) plans to eliminate HCV globally by 2030. To achieve this goal, it has been recognized that specific strategies are needed to facilitate care and treatment for Indigenous peoples, as outlined in the WHO’s 2017 Global Hepatitis report [2].

Globally, Indigenous peoples face a disproportionate burden of disease and poorer overall health status compared to their non-Indigenous counterparts [7,8,9,10,11]. These disparities are often attributed to the social determinants of health, which reflect the systematic “social, political, historical, economic and environmental factors, accumulated during a lifetime and transferred across multiple generations” [7], including institutionalized discrimination, racism, family separation, cultural genocide, land, and reconciliation factors that impact health status [10,12,13,14,15,16,17,18]. These discrepancies can be exacerbated in rural and remote areas; for example, Aboriginal and Torres Strait Islander peoples living remotely in Australia are 4.3 times more likely to experience a potentially preventable hospitalization than those living in major cities [19]. They are also significantly more likely to be diagnosed with HCV, but access to HCV treatment by Aboriginal peoples in remote communities is halved compared to others residing in urban areas [20]. Poor access to health services for Aboriginal and Torres Strait Islander peoples living in rural and remote areas is well recognized, often due to lack of culturally appropriate services, distance to healthcare services and lack of affordable transport, risk of staying away from home to receive treatment without family support, increased financial costs, and perceived lack of confidentiality [21].

In rural and remote settings, ensuring access to primary healthcare is widely accepted as key to improving health outcomes [22], and is essential in all components of the HCV treatment cascade, including medication access, commencement, adherence, and completion. Aboriginal Community-Controlled Health Services (ACCHS) are a major provider of primary healthcare for Aboriginal and Torres Strait Islander peoples [23], providing equitable access to culturally appropriate and holistic care [24], and are essential in facilitating HCV treatment in rural and remote areas.

A disproportionate burden of disease affects Indigenous populations, and HCV rates are increasing despite the global target for eradication [25]. Higher notification rates of HCV have been reported in the Indigenous peoples of Canada, at 4.7 times higher compared to the non-Indigenous population [26,27,28]. Similarly, the incidence of HCV among the Indigenous peoples of America (American Indian peoples and Alaskan Native peoples) is reported to be 2.8 times higher and continues to increase, while the rate of HCV amongst the non-Indigenous population remains stable [29,30]. In Australia, the most recent data available demonstrate that the notification rate for new HCV diagnoses for Aboriginal and Torres Strait Islander peoples increased by 15% between 2013 and 2017, while the rate in the non-Indigenous population decreased by 12% [31], a concerning statistic in a time of planned global eradication. New HCV infections within the teenage Aboriginal and Torres Strait Islander male population (aged 15–19) were nine times higher than rates in the non-Indigenous population in the same age group [25]. Improving healthcare access through innovation, expansion, and increased appropriateness of existing healthcare services can reduce the multiple barriers to accessing healthcare among Aboriginal and Torres Strait Island peoples, particularly in rural and remote areas. This is a priority area of the Fifth National HCV Strategy 2018–2022 [32] and the Fifth National Aboriginal and Torres Strait Islander STI and BBV Strategy 2018–2022 [33].

Before 2011, interferon-based therapies such as combined pegylated interferon and ribavirin were the mainstay treatments for HCV [34,35], notorious for their adverse side effects and low efficacy [36]. They have since been replaced by direct acting antivirals (DAAs), which offer high cure rates, shorter treatment durations, and significantly reduced adverse side effects [37]. In 2016, DAAs were listed on the Australian Pharmaceutical Benefits Scheme (PBS) [38,39], affording Australians living with HCV a safe, effective, and affordable treatment and the possibility of HCV eradication [40,41,42]. In the absence of specific data related to the number of Aboriginal and Torres Strait Islander peoples who have accessed DAAs, the first national report on progress towards the elimination of HCV in Australia suggested that a scale-up of HCV diagnosis and treatment will be required to reach priority populations, including Aboriginal and Torres Strait Islander peoples, at the same rate as the wider community [43], and this will involve all levels of the healthcare system, particularly within primary healthcare. This paper aimed to explore the barriers and facilitators to HCV treatment for Aboriginal peoples in rural South Australia (SA) from a healthcare provider perspective.

## 2. Methods

This study was conducted in two phases. Phase 1 was a qualitative systematic review (PROSPERO: CRD42014014694) examining the barriers and enablers to diagnosis and treatment amongst Indigenous peoples living with HCV globally. This knowledge provided the basis for Phase 2, a qualitative study exploring the perceived barriers and enablers to providing HCV care to Aboriginal rural and remote communities in SA from the healthcare providers’ perspective. The two phases were integrated iteratively at the analysis stage using the ‘following a thread’ [44,45] methodology.

### 2.1. Systematic Review Methods

#### 2.1.1. Inclusion Criteria

The systematic review was conducted as per the Johanna Briggs Institute (JBI) guidelines for qualitative data [46] and the Preferred Reporting Items for Systematic Reviews and Meta-Analyses (PRISMA) statement [47]. Inclusion criteria were international peer-reviewed literature that discussed barriers and enablers to Indigenous peoples with HCV diagnosis/undergoing treatment, written in English and published after 2011. The index year of 2011 was chosen because this was when the two first-generation DAA agents were approved for use [42,48].

#### 2.1.2. Population, Context, Phenomenon of Interest

The term “Indigenous peoples” was respectfully adopted for the review as this was the most frequently used term within an international context to include all ethnic groups who are the original inhabitants of a given region [49]. Nonetheless, we acknowledge that in some countries and communities worldwide, other terms may be preferred. Articles describing the experience of Indigenous peoples living with HCV and the experience of healthcare professionals, providers, and clinicians in treating Indigenous peoples with HCV were included to assess the perspective of both clients and providers (supply side) within the treatment cascade. We have used the term “client” as synonymous with “customer” and “patient”, as “client” appeared to be the preferred term in the literature.

#### 2.1.3. Search Strategy and Study Selection

Searches of CINAHL, MEDLINE, ProQuest, Scopus, Web of Science, Informit, and Emcare, and grey literature databases (Trove, Google Scholar, and Mednar), as well as relevant websites (World Health Organization, American Association for the Study of Liver Diseases, and European Association of the Study of the Liver), were conducted. The search strategy used is contained in Appendix A. On completion of the search, citations were collated and uploaded into Endnote X9 (Clarivate Analytics, Philadelphia, PA, USA), and duplicates were removed. Titles and abstracts were screened by EP and AC with random audits by DL and CB. The full text of potentially relevant studies was retrieved and assessed in detail against the inclusion criteria. Disagreements between the reviewers EP and DL were resolved through discussion with CB or JW. Consensus was reached on all included papers.

Eligible studies were critically appraised by EP, AC and DL using the JBI Critical Appraisal Checklist for Qualitative Research. There were no disagreements between reviewers with consensus on all included papers. Data were extracted from the included studies using the standardized JBI Data Extraction Form for Qualitative Research. The research findings were pooled using the JBI meta-aggregation approach. This involved the aggregation of findings into categories, followed by the grouping of these categories to form a synthesized finding.

### 2.2. Qualitative Research Methods

#### 2.2.1. Study Design, Sample, and Recruitment

A qualitative descriptive design [50,51] was selected to gather open and rich responses of the experiences of health service providers in rural and regional SA to understand the process and perspective of stakeholders involved in the treatment cascade. The research team contacted executive staff from six rural and regional ACCHS in SA, and facilitated recruitment of health service staff upon providing endorsement. Eligible participants were aged 18 years or older, working in the ACCHS, and had knowledge and experience of HCV management within the Aboriginal and Torres Strait Islander community. Written informed consent from participants was obtained.

#### 2.2.2. Data Collection

Semi-structured focus groups were conducted on-site at each of the participating ACCHS between June and August 2018. Each focus group included 4–8 participants and was conducted by two researchers; EP led the focus group while DL or CB observed and took written notes. The research team recognized that direct questioning was not the most effective method for information gathering in this setting, and instead opted to use storytelling as a data collection method [52]. This culturally appropriate approach has been utilized in similar studies [53], allowing the discussions to be relaxed, where participants can lead conversations and contribute freely whilst still being deliberate and relevant to the research topic.

#### 2.2.3. Data Analysis

Focus group audio-recordings were transcribed verbatim by EP and de-identified. Consultation with an Aboriginal mentor (DC) was used to interpret underlying cultural context, and the transcripts were analysed and thematically organized using the process outlined by Braun and Clarke [54] to identify factors relevant to the research questions. The data were systematically coded across the entire data set before collating into potential themes. Consistent with other research in this area [55,56], further analysis was undertaken using the client/provider/system structure. Key quotes that contributed to the thematic analysis can be found in Appendix B. Although this study only involved healthcare service providers, the patient/client tier was included by examining barriers for Aboriginal and Torres Strait Islander peoples living with HCV, as perceived by healthcare providers.

The Lincoln and Guba Trustworthiness Criteria were utilized to ensure the integrity of the data due to the subjective nature of qualitative research [57]. This addressed credibility (by member checking and cultural validation), transferability (by including different regional, rural, and remote settings in South Australia during data collection), dependability (through audit trails whereby a reflective journal was kept throughout the data collection process detailing methodological decisions and personal reflection), and confirmability (which was met once credibility, transferability, and dependability were established [57].

### 2.3. Integration of Data

The results from both phases were integrated iteratively employing the ‘following a thread’ methodology previously described [44]. For instance, this inductive-led framework enabled the barriers identified by the international literature to be elaborated on by the rural and regional South Australia experiences to generate a multi-faceted understanding of how the HCV treatment cascade may be improved for Aboriginal peoples with HCV living in rural and regional SA. As many Indigenous communities view health and wellbeing as a holistic concept, an ecological approach that included factors on individual, relationship, community, societal, and culturally distinct levels was, therefore, used to organize the data [17]. Group discussion among the authors finalized the threads.

## 3. Results

### 3.1. Systematic Review Results

The search and screening process outcomes are documented in Figure 1; sixteen papers were selected for final inclusion. The key characteristics of the papers are summarized in Table 1. All papers were published between 2012 and 2020 and originated from Australia (57%), Canada (31%), and the USA (12%). The systematic review identified five key themes surrounding HCV treatment for Indigenous populations within a multi-national context, subsequently informing focus group themes for the qualitative study in Phase 2.

### 3.2. Qualitative Research

Thirty-four staff from six regional and remote ACCHS in SA participated in the qualitative research phase; the majority were female (82%), and 44% were Aboriginal health workers (AHW). See Table 2 for a full breakdown of demographics. The sessions ranged in length from 24 min to 63 min, with an average length of 43 min.

### 3.3. Thematic Analysis Integrating Phase 1 and 2

#### 3.3.1. Importance of HCV Education

Across the published literature, clinician and client knowledge of HCV and how this education is provided has significantly impacted diagnosis, engagement in care, treatment commencement, and treatment completion in three main ways. Firstly, many health workers and other clinical staff reported minimal knowledge of HCV and inadequate training [58,67]. Secondly, resources were often unsuitable for clients with lower levels of literacy [61,64], and clinicians generally did not confirm clients’ understanding [61,64]. A relaxed and culturally appropriate environment [64,67], integrating traditional norms and knowledge into resources and information [58], as well as education projects incorporating culturally based programs [58,67] were essential factors for promoting engagement. In one study, most of those who attended these education sessions subsequently participated in the screening, of which approximately half also returned for test results [67].

Finally, a lack of client knowledge about HCV was recognized as a barrier to accessing care. Misconceptions about HCV prevention, transmission, and treatment were described in the literature [61,67,72], suggesting disempowerment leads to confusion, fear, and shame when receiving an HCV diagnosis [61]. A positive association between increased HCV knowledge and the likelihood of treatment uptake was identified secondary to client empowerment [67], which also positively influenced lifestyle changes since diagnosis and future HCV treatment intentions [66], and decreased the length of time accessing medical support such as a family doctor or specialist.

The project’s qualitative research component validated this theme within regional and remote Aboriginal communities in SA. Health service providers working in this setting identified that both behavioural risk factors and limited knowledge regarding HCV transmission could mean that people are unaware they are at risk of HCV. In contrast, the sixth health service indicated that people would be aware of HCV and some risk factors, such as intravenous drug use (IVDU), but may not be aware of other modes of transmission. All participants agreed that there was limited knowledge of the new DAA medications for Aboriginal peoples at risk of HCV, and, therefore, clients might not know that HCV is curable. Hesitancy to adopt the new DAAs was also an issue due to personal or community experience of the severe side effects of previous interferon-based therapies [AHWs, ACCHS #3 and #4].

Furthermore, study participants suggested that lower levels of education and poorer engagement with health services were barriers to HCV treatment within regional and remote Aboriginal communities. Health services were the locations where clients were most likely to notice information about DAAs. Poor engagement with these services could contribute to the lack of knowledge in the rural community. This highlights the importance for HCV resources to be visual, in the local language and in English, and broadly advertised throughout communities, not just in health services. Study participants reported a lack of appropriate resources to give to diagnosed clients [RN, ACCHS #4] and difficulties conveying complex information such as medication adherence, differing genotypes, and risk of re-infection to their clients.

Reflecting on the theme present in the literature, clinician education was also raised as an important issue for regional and remote healthcare workers working in Aboriginal health. Many study participants mentioned a significant lack of HCV-specific education for staff [RNs, ACCHS #1 and #2], and those who felt they had a good knowledge of HCV reported learning through self-education and via pharmaceutical representatives rather than through formal education pathways.

#### 3.3.2. Competing Demands and Health Priorities for HCV Treatment

Internationally, Indigenous peoples often face competing demands and health priorities that can influence HCV treatment commencement and completion. Obligations such as family relationships and cultural commitments were an important barrier to treatment commencement, as the physical effects of the medication may impact their ability to look after others [61]. Additionally, significant cultural obligations surrounding death within Indigenous communities are often prioritized before individual health [67], leading to poor adherence to medication regimens and prolonged time away, resulting in missed follow-up. However, familial connections were also seen as a motivator to engage with health services and access treatment [68].

Many clients had other co-morbidities, and a number of papers reported that Indigenous peoples were acutely aware of the higher burden of disease experienced by their community, in which an HCV diagnosis could be overwhelming, preventing engagement with health services and treatment [71]. Conversely, fear of the health consequences of untreated HCV provided motivation for some clients to be treated, especially for those who had experience with liver disease within their families [68].

In regional and remote Aboriginal communities, the findings from the qualitative research phase emphasized that cultural obligations could delay treatment uptake and contribute to poor medication adherence and loss of follow-up. Healthcare workers confirmed these obligations might take precedence over one’s health and influence the ability to engage in care, for example, family responsibilities in looking after extended family or the increased burden of illness and a high number of funerals [GP, ACCHS #3]. “Sorry business” in Aboriginal communities was more important for clients to attend and a higher priority than health appointments. Study participants suggested that a counterintuitive delaying of treatment until social and emotional needs have been met could contribute to improved outcomes.

Health for Aboriginal peoples is about physical wellbeing and emotional and spiritual health [74]. The asymptomatic nature of early HCV infection may impact treatment uptake and completion, as clients often do not feel acutely unwell [GP, ACCHS #3]. However, study participants also highlighted that HCV could be considered less of a priority when other comorbidities and Aboriginal peoples’ social and emotional wellbeing are also affected. The accumulative nature of problems can become overwhelming [RN, ACCHS #1]. Therefore, an approach that focuses on holistic healthcare, which supports both emotional and spiritual health as well as physical, could result in better engagement in health services within regional and remote communities [AHW, ACCHS #1].

#### 3.3.3. Care Delivery and Client Experience Impact HCV Treatment Outcomes

Access to healthcare is influenced by socio-environmental factors, care delivery, and the overall health service experience, which can impact HCV diagnosis, and treatment commencement and completion in Indigenous populations. Low socioeconomic status, limited transport, living rurally, and homelessness were significant barriers to accessing HCV healthcare in Indigenous people [59,61,63]. Building strong community connections, having flexible walk-in appointments, short wait times, child-friendly waiting areas, and incentive peer referral programs effectively increased access to HCV treatment [67].

Not receiving culturally appropriate care could also decrease treatment commencement and completion [75]. Other barriers include feelings of mistrust towards mainstream health services and clinicians as a result of colonization and lack of cultural awareness [64,65,71], as well as a “medicalized” culture within the hospital system [68]. Having an Indigenous health worker involved was seen as beneficial by building strong connections with the community and reinforcing the notion that the program was run “by Aboriginal people, for Aboriginal people” [67]. Other ways of providing culturally appropriate care included pre-employment cultural health training for non-Indigenous staff [67], peer support, and weekly visits from ACCHS staff [61]. Overall, delivering care in a friendly, non-judgmental, and culturally sensitive environment made clients feel more comfortable, and they were more likely to return in the future [67,68].

In Phase 2 of this study, healthcare workers expressed concern regarding delivering culturally appropriate care within regional and remote Aboriginal communities. Clinicians were trained in the bio-medical aspect of health, but if the cultural aspect was not acknowledged, this could result in treatment disengagement [RN, ACCHS #4]. Study participants discussed the diversity amongst Aboriginal communities and the importance of understanding the local culture in which healthcare professionals were working, and if unsure, the importance of asking to avoid breaching cultural protocols [RN, ACCHS #4]. The role of gender was also an essential concept to understand, as some Aboriginal peoples may not be comfortable discussing their health with a healthcare worker of the opposite gender due to culturally sensitive topics. Study participants suggested that cultural induction training for staff and ongoing professional development led by an AHW could ensure the provision of culturally appropriate care.

#### 3.3.4. Internal Barriers to HCV Treatment Commencement and Completion

Barriers to accessing treatment included the typically long wait times involved with referrals to specialists [58,61]. A shorter wait time between diagnosis and specialist appointments could help reduce the distress associated with being diagnosed [61]. Not linking clients to services or not having frequent and regular specialist access was also identified as a barrier to HCV treatment, particularly in a rural setting [72]. Telehealth services within Indigenous communities can help overcome these barriers [72].

Medical side effects, past experiences, and eligibility also impacted successful HCV treatment. Clients were concerned that side effects would negatively impact their ability to provide for their families [61]. It is important to note that significant adverse side effects were generally reported in the context of interferon-based treatments, which reduced the willingness of clients to access treatment [68]. Eligibility criteria for treatment were reported as a barrier, with clients describing instances they were not eligible for treatment due to injected drug use, alcohol intake, and psychiatric conditions [58,59]. Administrative challenges such as requiring late-stage liver disease and documentation of extended periods of sobriety were also identified as significant barriers to treatment [72], as was the cost of medication for those uninsured in countries such as the USA [59].

Within the qualitative study phase, the transient nature of living was noted as a factor that must be considered for Aboriginal clients in regional and remote areas. Frequent moving from place to place was mentioned by study participants as a primary reason for the disruption to the continuity of care and poor medication adherence [RN, ACCHS #4]. Study participants felt that point-of-care testing (POCT) had been very beneficial within regional and remote Aboriginal communities, particularly with transient clients. It was suggested that there needed to be better partnerships between ACCHS and mainstream primary, secondary, and tertiary health services to ensure information was not lost and that there was continuity of care regardless of which health service the client attended. One focus group spoke about their relationship with the local prison in providing health summaries after discharge, which enabled follow-up with their clients, further illustrating the benefit of cross-provider solid relationships where strong rapport is built.

Another recognized barrier to care included health service access, difficulties with the availability of prescription medications such as DAAs, and the retention of healthcare workers in regional and remote areas. Some services indicated they relied on locum doctors, and felt this did not allow clients to build sufficient rapport with their clinicians. This jeopardized the aim of providing holistic care, as clients were less likely to discuss their HCV risks, diagnosis, or treatment without trust.

In relation to this, confidentiality was also seen as a key issue in highly mobile regional and remote Aboriginal communities, and study participants felt it would be a breach of confidentiality to disclose a client’s medical history or HCV status to their current community’s healthcare service in the location where they have moved to [RN, ACCHS #1]. A suggestion to help remedy this would be to notify the new health service that the client needed to be seen by a doctor without disclosing the nature of their complaint. Participants felt this notification would enable proactive healthcare within a transient population without breaking the client’s trust [RN, ACCHS #1]. Guaranteeing that client records were only viewed by authorized people within the health service could ensure confidentiality [RN, ACCHS #2].

#### 3.3.5. Overlapping Stigma, Discrimination, and Shame

Overlapping stigma, discrimination, and shame can prevent engagement in HCV care and lead to a loss of social support and cultural identity. Within the international literature, HCV-associated stigma was a significant barrier to seeking appropriate healthcare. Many clients reported concerns regarding how others would perceive them if they disclosed their HCV status [61,64,65,67]. Other barriers to treatment included general negative experiences of Indigenous peoples within mainstream health services [68], encountering discriminatory attitudes and treatment from others [65], and the impacts of stigma on accessing and providing family support [70]. Health service access and engagement significantly increased when the model of care was client-centred and free from stigma [67]. Recommendations for reducing HCV-related stigma included providing information about HCV to people in the community who did not necessarily have HCV to promote understanding and acceptance, and encourage engagement with health services [67]. While it was found that Indigenous health workers generally played a key role in navigating potential stigma in mainstream health services, the stigma associated with perceived current drug use if seen attending an ACCHS was a potential barrier to accessing services [61,65].

Cultural shame was reported as an important barrier to Indigenous peoples’ engagement in care and treatment as it prevented them from talking about HCV with healthcare professionals [68]. Clients often believed they would bring shame to themselves and their friends and family through their HCV status and, consequently, be alienated by their communities [65,68]. This greatly impacted clients’ cultural identity, with some describing it as the ‘broken spirit’ disease [60]. One paper reported that the shame of HCV was largely related to its implication of injecting drug use and the shame related to any form of health vulnerability [68]. In addition, clients reported that they often did not receive counselling in relation to their diagnosis, which led to further feelings of confusion and shame [61,64]. In contrast, some papers suggested that feeling connected to others experiencing similar stigma and having a strong sense of cultural identity and community attachment could help buffer feelings of shame, and were associated with positive health behaviours [62,65].

Extending this theme to a regional and remote context, the study participants reported that the negative cultural stigma associated with HCV diagnosis could elicit an emotional response of fear, confusion, and shame. Participants thought that clients would not want to disclose their HCV status, which could be understandably difficult in small communities where many of the health professionals working in the health service may be relatives of the client, possibly leading to embarrassment and avoidance of treatment due to shame [AHW, ACCHS #1]. By encouraging general community discussion about HCV, factual information could be promoted and awareness could be raised, which can help normalize the condition and reduce HCV-related stigma. Some study participants said they had heard of positive experiences using peer support workers, but this would need to be handled sensitively due to the cultural stigma.

The interviewed healthcare workers identified that due to entrenched racism, Aboriginal peoples were more likely to experience a higher degree of stigma and discrimination in general, and an HCV diagnosis could further compound this. For example, the strong association of HCV with IVDU could result in unwanted labels and stereotypes, acting as a further barrier to HCV treatment access due to the patients’ wish to not be labelled this way. For example, in a family court setting, a prior diagnosis of HCV could reflect poorly on the perceived parenting capabilities of Aboriginal families due to racial stereotypes and biases [AHP, ACCHS #2].

## 4. Discussion

To our knowledge, this is the first published research incorporating a systematic review to explicitly examine the barriers and enablers to diagnosis and treatment amongst rural Indigenous peoples living with HCV in Australia. The review revealed five key themes, which provided a basis for the subsequent qualitative study focusing on regional and remote Aboriginal communities in SA. The main findings of the study revealed that appropriate HCV knowledge and education, recognizing aspects of holistic care and health and cultural priorities, and providing better access to care in a culturally safe way, as well as logistical barriers and the effect of stigma and discrimination, continued to play key roles in how Indigenous peoples navigate the healthcare system and their decision to engage in HCV care in the era of DAAs.

Improved clinician and client knowledge of HCV by providing staff education and training and culturally appropriate resources to educate and empower clients on HCV transmission and the relatively new DAAs can significantly impact HCV diagnosis, engagement in care, and treatment for the Indigenous population [76,77,78]. An awareness of cultural and social obligations, such as the importance of extended family and cultural commitments, can lead to better follow-up and adherence to HCV medication when an informed and flexible healthcare approach providing holistic care is followed. In addition, ensuring adequate access to healthcare and continuity of care within transient populations and providing culturally safe care builds trust and rapport to facilitate engagement within health services. It was recognized that due to difficulty employing clinicians in rural and remote areas, non-resident, non-Indigenous clinicians frequently work within these communities [79,80], and locally delivered AHW cultural awareness training, as well as ongoing professional development training, could enable better engagement with the community and understanding of local Aboriginal culture when providing HCV-related healthcare. Confidentiality was a key factor for regional and remote Aboriginal communities, and a clear strategy to ensure privacy will aid in removing some barriers in this context.

HCV-related stigma and shame were identified as significant barriers both in the systematic review and the qualitative study, and were known to impact multiple aspects of care including prevention, management, and treatment [81]. The cultural notion of shame is a powerful emotion for Aboriginal and Torres Strait Islander peoples that results from the loss of extended self, and it profoundly affects healthcare outcomes [12]. It is important to note the overlapping stigma and shame associated with HCV, presumed stereotypes of IVDU, and cultural identity, and interventions to improve care and treatment must address these overlaps. This is imperative in rural and remote communities where strict anonymity is difficult and facilitating treatment outside of the residential area may be an option to reduce this barrier; however, shared care between healthcare providers in this context becomes even more critical. Health providers have a constructive role in de-stigmatizing HCV, and the provision of culturally appropriate care as well as education can increase understanding and dismantle HCV preconceptions, leading to more social acceptance, stronger social support, and increased engagement with healthcare.

Many developed countries such as Australia are fortunate to have access to highly effective DAA treatments, and if the global and national target to eliminate HCV by 2030 is to be realized, the remaining barriers to providing effective care to Aboriginal and Torres Strait Islander peoples in regional and rural communities must be overcome. In their service to their community, ACCHS are uniquely placed to identify these barriers and fulfill the role of curative treatment as well as preventative treatment. Clients access ACCHS for a variety of reasons, from chronic disease management to a forum for social interaction between community members, therefore combining the HCV treatment cascade within the general model of care and service may have the advantage of giving HCV clients the anonymity to come forward to be tested, diagnosed, and treated. The exploration of the perspective of ACCHS healthcare providers enables a deeper understanding of clients’ challenges and, therefore, gives essential information on how providers can facilitate increased DAA uptake and treatment completion. Whilst this research focuses on the latter part of the HCV treatment cascade, the findings may nonetheless also inform strategies for preventing HCV.

There are limitations to our findings. The included papers within the systematic review were exclusively from Australia, Canada, and the USA, despite our various attempts to search for a broad range of Indigenous population terminology. The systematic review findings may not be representative of all Indigenous populations, and this reduces the generalizability of the synthesis; however, previous research suggests that Indigenous peoples around the world face similar healthcare challenges [9], and so many of the recommendations are likely to be transferrable. A further limitation of the qualitative element of this paper was the inability to capture the clients’ perspective, which is an area for further research. ACCHS in urban areas (a key community at high risk of HCV infection [25]) and Torres Strait Islander communities were not represented in this research; however, this study does provide insight into key issues facing regional and remote Aboriginal communities in SA. This study was intended to be the first of a larger project that aims to understand how to optimize HCV care for Aboriginal and Torres Strait Islander peoples, and further research involving clients and community members is needed to give a voice to those living with or affected by HCV, especially in rural and remote areas. Additionally, the perspectives of policymakers, peak bodies, and health service providers are key for the future development and evaluation of appropriate health information resources.

## 5. Conclusions

The introduction of DAA medications offers the chance to completely eliminate HCV. However, there is a clear disparity in HCV outcomes within Indigenous populations, and in Australia, the rate of infections within Aboriginal and Torres Strait Islander communities is increasing. This study emphasizes the importance of education within healthcare for a disproportionally affected population and explores barriers at the individual, provider, and system levels, especially stigma and shame. Key implications for public health policy include emphasizing culturally appropriate HCV education for clients, the community, and health service providers. Continued efforts to facilitate the uptake of DAA medications for Aboriginal peoples in rural and remote areas should use a multifaceted approach to provide education to clinicians and the community, increasing HCV knowledge and cultural awareness, and aiding in reducing stigma and discrimination.

## Figures and Tables

**Figure 1 ijerph-20-04415-f001:**
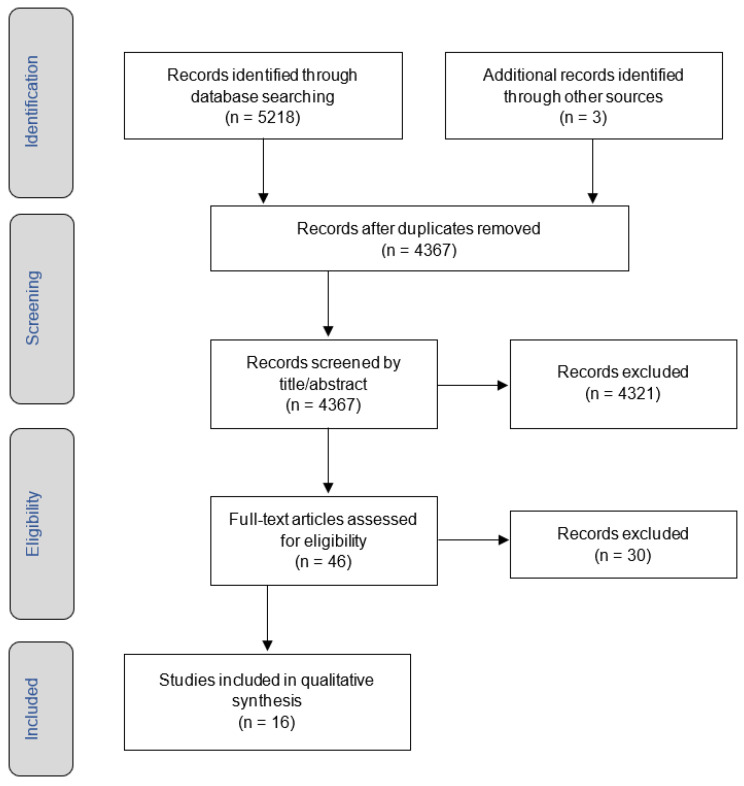
PRISMA search and study selection process [47].

**Table 1 ijerph-20-04415-t001:** Included papers in the systematic review.

Reference	Year	Country	Population of Interest	Data Collection Method	Topic	Sample Size
Livingston et al. [58]	2012	USA	Alaska Native and American Indian peoples	Retrospective chart review	Eligibility for treatment initiation and reasons for referral	146
Reilley et al. [59]	2014	USA	Alaska Native and American Indian peoples	Survey	Barriers to HCV screening and access to treatment	48
Hossain et al. [60]	2014	USA	Native American peoples	Retrospective chart review	Barriers to HCV treatment commencement and completion	22
Treloar et al. [61]	2015	Australia	Australian Aboriginal and Torres Strait Islander peoples	In-depth interviews	Experiences of and influences on decisions about diagnosis, care, and treatment	39
Brener et al. [62]	2015	Australia	Australian Aboriginal and Torres Strait Islander peoples	Survey	Effect of community attachment on health outcomes, lifestyle changes, and treatment intention	203
Parmar et al. [63]	2015	Canada	Canadian Aboriginal peoples	Retrospective chart review	HCV risk factors and barriers to engagement in care, treatment initiation, outcomes, and completion	55
Brener et al. [64]	2016	Australia	Australian Aboriginal and Torres Strait Islander peoples	Survey	Delivery of diagnosis, ongoing care, and barriers to treatment uptake	203
Treloar et al. [65]	2016	Australia	Australian Aboriginal and Torres Strait Islander peoples	In-depth interviews	Impact of stigma, shame, and historical trauma on living with HCV, diagnosis, care, and treatment	39
Wilson et al. [66]	2016	Australia	Australian Aboriginal and Torres Strait Islander peoples	Survey	Association between HCV knowledge and health behaviors	39
Treloar et al. [67]	2018	Australia	Australian Aboriginal and Torres Strait Islander peoples	Retrospective chart review	Evaluation of program for engagement in HCV education, screening, and care	55 and 13
Wallace et al. [68]	2018	Australia	Australian Aboriginal and Torres Strait Islander peoples	Semi-structured interviews	Barriers and enablers to treatment, and patient experience of treatment and cure	50
Gachupin et al. [69]	2018	USA	American Indian peoples	Analysis of HCV data and medical records	Tribal-based initiatives for addressing HCV, from screening and diagnosis to treatment	251
Lakhan et al. [70]	2019	Australia	Australian Aboriginal and Torres Strait Islander peoples	Retrospective chart review	Optimizing HCV care based on demographics and clinical characteristics	113
Pearce et al. [71]	2019	Canada	Canadian Aboriginal peoples	Semi-structured interviews	Supporting HCV treatment engagement and decolonization of care for Indigenous peoples impacted by substance use	55
Stephens et al. [72]	2019	USA	Alaska Native and American Indian peoples	Survey	HCV telehealth services for linkage to specialists	44
Rashidi et al. [73]	2020	Australia	Australian Aboriginal and Torres Strait Islander peoples	Semi-structured interviews and survey	Factors affecting HCV treatment intentions	10 and 123

**Table 2 ijerph-20-04415-t002:** Participants of the qualitative research (n = 34).

Demographic	Number	%
Male	5	15
Female	28	82
Non-binary	1	3
Total	34	100
Aboriginal Health Worker	15	44
Registered Nurse	10	29
General Practitioner	2	6
Allied Health Practitioners	5	15
Health Management Role	2	6
Total	34	100

## Data Availability

The data presented in this study are available on request from the corresponding author. The data are not publicly available due to the conditions of institutional ethics approval.

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
