# Peer review of "Barriers and Facilitators to Hepatitis C Virus (HCV) Treatment for Aboriginal and Torres Strait Islander Peoples in Rural South Australia: A Service Providers’ Perspective"

_ijerph, 2023, doi:10.3390/ijerph20054415_

Round 1

Reviewer 1 Report

Thank you for giving me the opportunity to review the manuscript "Barriers and facilitators to Hepatitis C Virus (HCV) treatment by Aboriginal and Torres Strait Islander Peoples in rural South Australia: a service providers’ perspective " submitted to the International Journal of Environmental Research and Public Health.

The study by Dr David and colleagues aimed to improve the care cascade for Aboriginal peoples with HCV living in rural and regional South Australia. The study adopted mixed-methods approach (systematic review of the barriers and enablers to diagnosis and treatment amongst Indigenous peoples living with HCV globally, and the actual barriers and enablers to providing HCV care to Aboriginal rural and remote communities in South Australia from the health care providers’ perspective). The researchers recommended strategies that can facilitate care and treatment for people in these areas.

The manuscript is well-written, the method is sound and the findings of the study have practical implications. Very interesting paper! Good job and thank you for sharing.

I would suggest the below minor suggestion:

The manuscript is written in a logical and consistent way; however, could you please summarize the main actual barriers to HCV treatment amongst Aboriginal people living with HCV listing them in order of significance at the end of the Discussion section or in the Conclusion section. 

Author Response

Dear Reviewer, thank you for the constructive feedback for improvement and positive affirmation. We have now incorporated the changes in the revised manuscript.

Reviewer 2 Report

Dear authors,

The title of Your work intrigued me very much to accept to do the review of your work and I really enjoyed reading it.

I have no remarks to this work, but I must disclaim that I am a basic scientist who also sometimes does systemic review with meta-analysis, so I am very familiar with that part of the work. However, the second part of the analysis done in the paper - qualitative research method and “following the thread” methodology is not something that I am familiar with in detail, only superficially, so I can't see if you missed something there and if something needs to be added.

All in all, the work is a great read for me with some new information and insights, and it once again showed how important education is, as well as the way one approaches and interacts with people/patients, while knowing and respecting one's cultural beliefs and heritage.

Author Response

Dear Reviewer, thank you for the positive affirmation and support. It is sincerely appreciated.

Reviewer 3 Report

Barriers and facilitators to Hepatitis C Virus (HCV) treatment by Aboriginal and Torres Strait Islander Peoples in rural South Australia: a service providers’ perspective

A brief summary

Hepatitis C virus (HCV) is one of the leading causes of liver-related morbidity and mortality worldwide, affecting more than 70 million people. Approximately 55% to 85% of infected people will develop chronic hepatitis C and 15% to 30% of this group will develop liver cirrhosis and hepatocellular carcinoma within the next 20-30 years.

Australia is an international leader in its move towards managing hepatitis C in primary care. Starting on the WHO's brave goals include: improving primary prevention (e.g. safe injections in healthcare facilities and for people who inject drugs), expanding hepatitis C screening and linkage to care, increasing access to antiviral treatment, and reducing the impact of the disease on the population (reducing the rate of hepatitis C), health care providers in disadvantaged areas of Australia have tried to adapt these goals to the environmental, social, cultural and individual conditions of the indigenous communities of southern Australia.

Indigenous peoples in South Australia often find it difficult to access adequate primary health care services. Ensuring access to health care for Indigenous peoples who often face a wide range of additional barriers, including experiences of discrimination and racism, can be complex. The uptake of direct-acting antiviral therapy among these rural communities will result in a reduction in deaths from hepatitis C-related liver problems and a reduction in the costs of treating these patients, thereby reducing pressure on national public health systems after 2030.

General comments

The main objective of the study was to highlight the means, but also the problems encountered in applying the treatment to the Aboriginal and Torres Strait Islander population in rural South Australia from the perspective of medical service providers in the DAA period. Aboriginal and Torres Strait Islanders account for approximately 3% of the Australian population. They have the poorest health, economic and social outcomes. In Australia, for the population in these rural areas, the most important provider of health services is the Aboriginal Community Health Service (ACCHS), which provides culturally appropriate care in particular and facilitates access to antiviral treatment for the population at risk. One of the biggest problems faced by health service providers is the limitation of diagnostic and therapeutic possibilities for patients with chronic hepatitis C in these communities, due to the harmful influence of geographical, social, political, historical and economic factors, accumulated during life and unfortunately passed on to future generations. I believe that mentioning the patient eligibility criteria for this study would have been necessary to increase the accuracy of the clinical study.

Specific comments

Introduction

In Australia, hepatitis C virus infection is a major public health problem due to the progression to chronicity, with a very high risk of liver cirrhosis and hepatocellular carcinoma. Since the 1990s the diagnosis of chronic hepatitis C has been submitted to the Australian National Communicable Disease Surveillance System which has reported an incidence and prevalence of the disease of 80%, particularly in the susceptible population (drug users). The incidence of this disease has decreased considerably as a result of the introduction of screening programs, but also of direct-acting antiviral therapy (DAA), starting in 2016. As a result, the present study addresses a topic of great scientific interest considering the increased rate of chronic hepatitis C, but also the lack of studies on the barriers encountered in ensuring the treatment of the disease, especially for the population from poor environments.

Material and methods

One of the strengths of the article is that the authors are based on two study models, one of the qualitative systematic review type, which sought to identify the barriers, but also the facilitators that contribute to the diagnosis and treatment of the disease at the level of the global population, and one of descriptive qualitative type, which aimed to identify the challenges and factors that provide support for the provision of care to the rural and remote community in South Australia from a health services perspective.

The combined use of two studies with different objectives gives the article originality and opens new horizons for research and implementation of screening programs for the diagnosis and initiation of DAA therapy, especially for the susceptible population. The systematic review focused on presenting the diagnostic and treatment issues described by other authors globally however, most studies came from Australia (57%) followed by Canada and the USA which identified five key themes that mainly addressed the treatment of the disease for the indigenous population at the multinational level but later also provided the themes for the descriptive study. Thus, the advantages of the study mainly come from the descriptive study because it only refers to the problems experienced by the population of South Australia. In order to better understand what the problems are and find their solution, the descriptive study included ACCHS members from the authors' target region who know and better understand these problems that need to be solved in view of the proposed WHO initiative. The information used in the article was selected from the specialized literature using the most suggestive studies that targeted the barriers and supporting factors for the indigenous population undergoing disease diagnosis and treatment published after 2011. The article was produced according to the Johanna Briggs Institute (JBI), and the information was selected with the help of PRISMA, which represents systems with highly reliable qualitative results. The materials used were obtained from internationally recognized databases, but also through the most well-known websites that provide information about viral C infection, such as the WHO, the American Association for the Study of Liver Diseases, and the European Association for the Study of the Liver. All these strengths give the study great scientific value.

Results                                                                                                          

Starting from the particularities of HCV infection in the rural indigenous population of southern Australia, in the absence of a vaccine to protect the population against HCV, the only possibility to eliminate the virus from rural communities remains the current modern antiviral treatment, based on direct antivirals. That's why healthcare providers have set a series of goals in recent years, which aim to eradicate the virus from rural indigenous populations in southern Australia by around 2030. The authors of this study tried to identify all the barriers and facilitators of direct antiviral treatment and minimize the effects of the barriers, while at the same time promoting the effects of the facilitators of antiviral treatment.

The main issues that the authors discuss in this article, with the aim of eliminating the barriers to the diagnosis and treatment of hepatitis C, among disadvantaged populations, in southern Australia are:

1. Cancellation of some barriers related to the knowledge of the disease and its treatment in the indigenous population was the first topic addressed by the authors of this study. The providers of medical services took into account the low levels of education of the clients and were able to adapt the speech to the level of preparation of the clients and to their language (dialect) this, avoiding a series of barriers that could have delayed the introduction of antiviral treatment directly in the indigenous population. Also, the medical personnel, who might have lacked information about the disease, accepted the information more easily through pharmaceutical representatives than through formal education.

2. Starting from international information regarding "Concurrent requests and health priorities for HCV treatment" in indigenous populations, several barriers to starting and completing HCV treatment were identified. Thus, family obligations, cultural obligations, and the presence of comorbidities can negatively influence antiviral treatment. Therefore, the knowledge of these barriers could lead to the holistic care of the patient's health, taking into account also emotional and spiritual health.

3. Improving social and environmental factors, improving the quality of medical services, and changing patients' perception of the side effects of direct antiviral medication will allow increasing access of indigenous people to specialized health services and finally improve the diagnosis and treatment of all patients with chronic viral hepatitis C.

4. The internal barriers to starting and completing HCV treatment refer to all the specific factors (e.g. long waiting times for addressing a specialist doctor, lack of regular access to a specialist doctor, difficulties in obtaining medicines from pharmacies) in the inhabited territory by the aboriginal populations, which limits the access of these people to specialized medical services. These internal barriers could be overcome by the implementation of a new health service in which the following should be pursued: the use of telemedicine within the health services of indigenous communities, the increase in the level of education regarding chronic hepatitis C of the health personnel, as well as of the clients, testing patients in a medical care point specially designed within the aboriginal community in which the medical personnel is recruited from within these rural populations, as well as the examination of the patient by the specialist doctor without returning the causes of their presentation to the doctor.

5. Psychological and behavioral barriers prevent many HCV-infected individuals from initiating or engaging in HCV treatment. These findings indicate that all HCV individuals should be counseled and encouraged to participate in educational programs at the time of diagnosis to reduce unnecessary behavioral changes and perceptions of stigma to improve their quality of life. As it permeates even the healthcare environment, physicians and other care providers should be aware of the existence and impact of such negative stereotypes.

Discussions

The study is particularly valuable because the practical application of solutions to increase the population's access to direct antiviral treatment, which addresses both medical service providers and local communities and the indigenous population, will lead to a drastic reduction in the number of sick people. Indigenous health care services must overcome both the social and cultural barriers to health care that prevent Indigenous people from accessing health care. The findings of this synthesis also suggest that by approaching the disease holistically, under all its aspects (epidemiological, diagnostic, and therapeutic management), it will be possible to achieve a reduction in the transmission of the hepatitis C virus, and in the future the emergence of hope for the eradication of this infection in these rural communities as well, disadvantaged, populated by indigenous people, from southern Australia.

Author Response

Dear Reviewer, thank you for the detailed analysis of the manuscript, it is sincerely appreciated. The authors are passionate about reducing the inequity between First Nation Peoples' health especially in this sensitive topic. We will incorporate the Reviewer's insight into our future research and clinical work. Again, we thank the Reviewer for your insightful and positive affirmation.

Reviewer 4 Report

I appreciate the opportunity to review this interesting research manuscript Title: Barriers and facilitators to Hepatitis C Virus (HCV) treatment by Aboriginal and Torres Strait Islander Peoples in 3 rural South Australia: a service providers’ perspective. This is an excellent and interesting study dealing with significant technical matters. the authors have collected a unique dataset. The paper is generally well-written and structured, I find no fault whatsoever with the methods, data analysis, or conclusions. The work, as with all work coming from this particular group, is fundamentally sound. My comments here are concerned solely with the organization of the manuscript. Consideration of these points will, I believe, lead to an improved report that better illustrates the key concepts and conclusions.

1.      There are numerous strengths to this study, including its diverse sample and well-informed hypotheses.

2.      The innovations of this manuscript are limited. Most of the results have already been described in some review papers

3.      the term “Indigenous peoples” is very difficult to understand for a layman my suggestion is to  express it in simple and straight sentences with aspect to readers’ aspect

4.      Abstract: Please focus the abstract on your study and your results. In particular, the last two sentences are vague. I would prefer to see some data on HCV association with HCV and DAA, in the Australian population rather than a description of “where to go next”. More generally, I suggest focusing the manuscript on the scientific results rather than on public health awareness.

5.      Authors must explain the terms of approaches of Lincoln and Guba’s Trustworthiness Criteria in the discussion

6.      Authors must explain the inclusion and exclusion criteria of the data integrity parameters

7.      Did the authors use any questionnaire or any other tool to analyze the knowledge and concept of people about HCV

8.      The authors have not commented upon the different incidence of HCV in various countries in the current year 2022 and their impact on the results of this study. That remains an important factor to consider while looking at countries with a large number of publications. The research trend would be a reflection of the burden of HCV in that particular country. 

9.      Much more detailed reasons are required on how acute liver failure (ALF) occurs due to HCV in this cohort of Australia

10.   Page 4: line 135- only one reference is not sufficient for this statement

11.   Page 4 lines 163- 165- the sentence is very hard to translate.

12.   What were the inclusion and exclusion criteria why and on which basis authors included all other Aboriginal and Torres Strait Islanders and exclude their neighboring population from the study

13.   There was no mention of the limitations of the study, one of which is the high dropout rate. Also, mention how your results compare to (reference given to author) another study that was published very recently

14.   How this study could prevent HCV  in Australia Please elaborate and discuss.

Author Response

Dear Reviewer, thank you for the constructive feedback and the opportunity to improve the manuscript.

We have adopted the term "Indigenous peoples" respectfully in concordance with the UN charter and have defined this accordingly in section 2.1.2. We have also intentionally separated Australian Aboriginals from Torres Strait Islanders in the qualitative phase of the study. Thank you for alerting us to this nuance.

We have also elaborated on Lincoln and Guba qual research rigour in the revised manuscript.

We have kept the inclusion criteria for the systematic review broad: (i) published after 2011 (when DAA was approved), (ii) written in English, (iii) peer-reviewed literature which discussed barriers and enablers to Indigenous Peoples with HCV. The inclusion criteria for the qualitative study were rural and remote South Australian health service providers whom had experience and or knowledge of HCV management in the Indigenous community.

We did not directly interact with the HCV patients in this study and have acknowledged this as a limitation of this study. Thus, we do not have a drop-out rate per sec for HCV patients, nor inclusion criteria for patients.

Thank you for alerting to the liver failure, we have amended this sentence accordingly.

We have added more relevant references as recommended and redrafted the sentences with regard to JBI meta-aggregation.

We have revised the manuscript to focus on the prevention of HCV in Australia as suggested.

Thank you for the advice.

Reviewer 5 Report

Lim et al. assessed the barriers and facilitations to HCV in indigenous peoples living in south Australia and provided the five key determinants highly associated with the care of HCV in these peoples.

1.     The introduction was tedious. Please trim the wordings to make it more succinct.

2.     Sections 2.1.3 and 2.1.4 were duplicated, however, the contents were different. Please clarify it.

3.     Lots of grammatical and wordings errors were present in the whole manuscript, Hepatitis C virus, direct acting antivirals, indigenous, aboriginal etc… should be shown in lower cases. Line 36: “HCV is highly stigmatized condition” was confusing and should be “HCV infection is a highly stigmatized condition”. Please proofread the whole manuscript with care.

Author Response

Dear Reviewer,

Thank you for the constructive comments to improve the manuscript for the wider readership of the Journal. The manuscript has now been independently audited for its written expression. 

We have now removed the duplication in 2.1.3 and 2.1.4 in the revised manuscript. Thank you for the insightful comment.